# Analysis of Risk Factors of Feather Pecking Injurious Behavior in Experimentally Raised Yangzhou Goslings in China

**DOI:** 10.3390/ani15050616

**Published:** 2025-02-20

**Authors:** Mingfeng Wang, Guoyao Wang, Wang Gu, Zhengfeng Cao, Yu Zhang, Yang Zhang, Qi Xu, Guohong Chen, Yang Chen

**Affiliations:** 1College of Animal Science and Technology, Yangzhou University, Yangzhou 225009, China; 18361310771@163.com (M.W.); wgy15050786372@163.com (G.W.); dx120210144@stu.yzu.edu.cn (W.G.); caozhengfeng@yzu.edu.cn (Z.C.); yuzhang@yzu.edu.cn (Y.Z.); zyang@yzu.edu.cn (Y.Z.); xuqi@yzu.edu.cn (Q.X.); ghchen2019@yzu.edu.cn (G.C.); 2Joint International Research Laboratory of Agriculture and Agri-Product Safety, The Ministry of Education of China, Yangzhou University, Yangzhou 225009, China; 3Key Laboratory for Evaluation and Utilization of Poultry Genetic Resources of Ministry of Agriculture and Rural Affairs, Yangzhou University, Yangzhou 225009, China

**Keywords:** feather pecking behavior, gosling, rearing environment, skin damage

## Abstract

Feather pecking (FP) can lead to severe feather damage and feather loss. Despite extensive research on feather pecking behavior over the past decades, the motivation for feather pecking remains unclear and is one of the challenges faced in poultry production. This study shows that goslings often pecked feathers during 4–5 days of age, most frequently directed at the back. Lower stocking density, higher population uniformity, and the provision of enrichment can reduce the prevalence of feather pecking. These results help to develop effective management and prevention strategies to reduce the negative effects of pecking behavior on goose health and performance.

## 1. Introduction

Feather pecking, a behavior in which goslings peck each other on an area or attack each other to injure them, poses a significant economic and welfare challenge in poultry production worldwide [1,2]. Feather pecking differs from aggressive pecking in that it occurs in a non-aggressive state [3,4]. Geese will peck their feathers or each other’s feathers, resulting in thinning or beating feathers on the back or tail. In mild cases, only the feather branches are pecked while the roots remain in the skin, and the injured geese do not react strongly [5]. Severe pecking can cause extensive feather and skin damage. Poultry that experience feather pecking exhibit impaired plumage conditions but also have significantly reduced production performance, which is particularly harmful in the case of goslings [3]. While certain information about the feather-pecking behaviors of chickens and ducks has been presented in references [6,7,8], the feather-pecking behavior of goslings still lacks comprehensive demonstration. As of now, there is no reasonable and effective management plan to stop the feather-pecking problem [9]. Although previous research has investigated the effects related to injurious behaviors and group pressure during the welfare assessment process [10], there is little information on feather-pecking behavior and how often it occurs. In addition, there does not appear to be any information on the body parts that are most frequently damaged when goslings are pecking and the factors that affect them.

Feather pecking in geese, a behavior observed across all ages but particularly acute among goslings, exhibits rapid contagion within flocks. Research suggests that feather pecking is influenced by a multitude of factors, including environmental conditions, early life experiences, and genetic predisposition [11]. High-density rearing environments, high temperatures, and humidity have been identified as risk factors for feather pecking, particularly during the brooding period when poultry are most susceptible [7]. The early life experiences of poultry play a crucial role in shaping their behavioral traits, including feather pecking. Studies have demonstrated that management practices during the early stages of development can influence the likelihood of feather pecking later in life [12]. Feather pecking behavior is heritable, suggesting that genetic screening and selection may help reduce its occurrence in future generations [13]. Crucially, once feather pecking develops in a poultry population, it can be difficult to control, and if behavioral traits associated with feather pecking can be identified and measured when the birds are young, they will provide predictors of feather pecking behavior [14].

Since feather pecking is difficult to stop once it has started, the focus should be prevention [15]. By identifying risk factors at critical stages early in life, breeders can reduce the likelihood of feather pecking behavior developing into a widespread problem [5]. To investigate the causes of feather pecking behavior in goslings, this study compared the differences in feather pecking behavior of Yangzhou goose goslings under different rearing conditions by using the observation record method and sampled HE sections of skin tissues on the backs of goslings with intact back feathers and those with featherless backs due to feather pecking to observe the differences. This study aimed to reveal the differences in feather pecking behavior of goose goslings under different environments and the mitigation measures, to provide a certain theoretical basis for scientific selection and breeding, and to improve the productivity and economic benefits of waterfowl.

## 2. Materials and Methods

### 2.1. Animals Feeding Management, Experimental Design and Facilities

All animal experiments were approved by the Animal Care and Use Committee of the Yangzhou University. Goslings were raised in a brooding housing system in Tiange Goose Industry Co., Ltd. in Yangzhou, China with normal feeding management. The average temperature was controlled at about 28 °C, and the average humidity was controlled at about 65%. The basal diet composition and nutrient level of gosling were consistent with the previous study [16]. Select 234 two-day-old Yangzhou goslings and place them under different environmental conditions. They will be raised in a brooding house feeding system equipped with drinkers and feeders, using tap water as the source of drinking water, and the goslings can feed freely. Feed was provided daily at 8:00, 14:00, and 20:00, and the immunization program and other feeding management programs were carried out according to the routine of meat geese production.

The experiment was set up with three replicates, and the strategy for a single-group experiment is shown in Figure 1. Goslings were reared using 0.25 m × 0.25 m × 0.25 m pens and different groups of experimental geese. Three infrared video cameras (Jindun, 1080p, resolution: 1920 × 1080, frame Rate: 25 frame/s, Nanjing, China) were set up above water surfaces and land areas in the pen of each group. Hikvision digital recorder (1080p, 32 channels, resolution: 1920 × 1080, 2.4 TB, Shenzhen, China) was used for recording the videos. The camera was placed at a height of 3 m above the ground to record the full pen floor area. Except for the treatment groups, the goslings had a male-to-female ratio of 1:2 to 1:3 and body weight of 80–90 g. From 3 days of age, we began to observe the geese’s feeding, recorded the data from 3 to 10 days old, and observed them from 9:00 to 11:00 and 15:00 to 17:00 every day of age. To ensure the uniform growth of goslings, they were provided with 23–24 h of lighting time from 3 to 7 days of age. After 8 days of age, gradually transition from 24 h lighting to using only natural light. The sampling method proposed by Altmann [17] was adopted. The data were expressed as the number of occurrences within the observation time. If a behavior occurs continuously within 5 s, it is counted as one observation result. Behavioral characteristics included feather pecking, lying down, feeding, drinking, grooming, and cage pecking. Modifications were made to identify behavioral categories based on the Zepp [6] study, and the specific behavioral parameters are shown in Table 1. Each experimental group was divided as follows:(1)Thirty goslings were allocated into four distinct groups based on varying densities: Density 12 with 12 goslings, Density 9 with 9 goslings, Density 6 with 6 goslings, and Density 3 with 3 goslings, each group enjoying unrestricted access to food and water.(2)Sixteen goslings were divided into two groups: a flat-rearing group and a net-bed-rearing group, with eight goslings in each group. The net bed is a rearing platform constructed with plastic that has appropriately sized mesh holes and is elevated to a certain height above the ground. Flat rearing utilizes a cement floor on which straw bedding is spread.(3)Sixteen goslings were divided into two groups, each consisting of eight individuals. Comparable weight group goslings featured all maintaining a comparable body weight, such as Control. The mixed-weight group had four goslings weighing 110 g or more and four goslings weighing less than 80 g.(4)Sixteen goslings were divided into two groups of eight each. One group was supplemented with a pecking substance, grass clippings, and the other group was left untreated as a control. When the goslings reached 5 days of age, the grass clippings were supplemented with feed in the afternoon of each day.

### 2.2. Feather Scoring of Goslings

The total feather scores of all the goslings from different experimental groups were statistically analyzed, and the scoring criteria and reference charts are shown in Table 2, based on a modification of the Gilani [7] study criteria.

### 2.3. Degree of Feather Pecking in Goslings

The categories of feather pecking include gentle feather pecking (GFP), severe feather pecking (SFP), and aggressive feather pecking (AFP), as outlined in Table 3, which was based on the research criteria of Zepp [6] and Dong [8] with modifications. The feather-pecking behavior hierarchy exhibited by goslings under diverse conditions, encompassing variations in rearing density, sex, humidity levels, size, and the inclusion of grass segments as a supplementary element, is currently being assessed.

### 2.4. Preparation and Staining of Skin Sections

Four individuals that were severely pecked, resulting in missing back feathers, and four individuals with intact back feathers that were not pecked were selected from the reared goslings, and skin tissue samples were collected from their backs, then paraffin sections were prepared and stained with hematoxylin and eosin (HE). The stained sections were photographed under a microscope for archiving. Afterwards, the obtained images were analyzed. The epidermal thickness, feather follicle cell diameter, and the number of feather follicle cells were observed and measured using the CaseViewer2.3 software.

### 2.5. Statistical Analysis

For the measurement of the frequency of behavior occurrence, the goslings randomly selected from each pen were regarded as the experimental units, representing their respective pens. Initially, Excel 2019 was used to organize the data and establish a database. Statistical analysis was performed using SPSS statistical software (SPSS 22, IBM SPSS software, Armonk, NY, USA). All data are tested for normal distribution by the K-S (Kolmogorov–Smirnov) test. The data that followed a normal distribution were subjected to a one-way ANOVA followed by Duncan’s multiple-range test to determine whether significant differences existed between the groups. The results were expressed as the mean ± standard deviation. For data that do not follow a normal distribution, the Kruskal—Wallis test is first used, followed by the Mann–Whitney test. The results are expressed as the median (interquartile range). *p* < 0.05 was considered statistically significant.

## 3. Results

### 3.1. Overall Behavioral Characteristics of Goslings During the Brooding Period

During the brooding period, goslings exhibited distinct behavioral patterns. They consistently followed other goslings, preferring to move in groups and engage in social interactions. Their innate curiosity drove them to explore and observe their surroundings, often leading to cage-pecking behavior, accounting for an average of 10.52% of all observed behaviors (Figure 2). This period of brood rearing was crucial for the goslings as they learned various behaviors, such as foraging and grooming, to prepare them for adult life. Notably, the average frequency of feather pecking behavior among the goslings was 14.87%. In addition, the damp feathers inadvertently triggered other goslings to peck at the affected area. As shown in Figure 3I–III, moist skin is more susceptible to feather pecking. Furthermore, when one gosling starts pecking at the feathers of a fellow gosling, it causes a collective feather pecking behavior in other goslings (Figure 3IV–VI).

### 3.2. Effect of Feather Pecking on Skin Damage and Epidermal Thickness in Goslings

As shown in the histological sections of Figure 4A,B, feather pecking resulted in notable damage to the epidermis and dermis of the skin, whereas skin that was not subject to feather pecking (unpecked) remained pristine. In terms of feather follicle cell development, the skin that was feather-pecked showed fewer feather follicle cells (Figure 4C,D). Feather damage was evident in the pecked goslings (Figure 4E,F). Further analysis showed that goslings with intact back feathers had significantly more follicular cells and diameters than those with bare backs (*p* < 0.05) and significantly less epidermal thickness than those with bare backs (*p* < 0.05) (Figure 5).

### 3.3. Analysis of Preferences and Degree Variations in Gosling Feather Pecking Behavior

As shown in Figure 6A, there is no significant difference in the frequency of gosling feather pecking behavior at different observation times (9:00 a.m. and 3:00 p.m.) (*p* > 0.05). In terms of feather pecking sites, goslings prefer pecking at the back (77.32%) and the head (11.14%) (Figure 6B). For day of age, there was no difference in gosling feather pecking frequency at different days of age (*p* > 0.05) (Figure 6C). Regarding the degree of feather pecking, 82.16% of the behaviors are gentle GFP behaviors, and 17.02% are severe feather pecking (SFP), while AFP behaviors account for only 0.82% and are concentrated mainly at 4 days old (Figure 6D). At this stage, goslings are establishing a hierarchy involving a strict pecking order, where higher-ranking geese have priority in feeding. Violations of this order can lead to warnings, punishment, and even fighting, which may explain the occurrence of AFP behaviors. After reaching 4–5 days old, once the hierarchy is established, AFP behaviors are significantly reduced.

### 3.4. Effects of Different Rearing Densities on Behavior and Feather Coverage of Goslings

The rearing density has a significant impact on the feather-pecking and cage-pecking behaviors of goslings. The frequency of cage pecking decreases as the density increases, with Density 12 being significantly lower than Density 3 (*p* < 0.05) (Figure 7A). Combining the relationship between feather pecking and age shown in Figure 7B, it is found that from 3 days old, goslings in Density 12 (*p* < 0.001) and Density 9 (*p* < 0.01) exhibited more frequent feather pecking behaviors compared to those in Density 6 and Density 3. This indicates a strong correlation between the frequency of feather pecking and the rearing density. After conversion, the appropriate rearing density for goslings is 24–36 per square meter of net bed, which corresponds to the range of Density 6 to Density 3.

In terms of differences in feather scores, it was found that the feather scores of the gosling populations in the high-density groups, Density 12 and Density 9, were significantly lower than those in the low-density groups, Density 6 (*p* < 0.01) and Density 3 (*p* < 0.05) (Table 4). This indicates that feather pecking behavior occurs more frequently under high-density rearing conditions, leading to reduced feather coverage.

### 3.5. Effects of Different Rearing Conditions on Gosling Behavior and Feather Coverage

The results showed that the frequency of feather pecking behavior in the Flat feeding group was significantly higher than in the Net-bed feeding group (*p* < 0.05). This difference may be due to the fact that in environments with higher relative humidity, the feathers of goslings are more likely to become damp, which encourages feather pecking. Additionally, increased moisture reduced the lying down behavior of goslings and increased grooming behavior (Figure 8A). As shown in Figure 8B, under conditions of Flat feeding, the feather pecking behavior of goslings gradually increased with age. Notably, the Flat feeding group exhibited the highest incidence of feather pecking, in stark contrast to the Net-bed feeding group, with approximately 70% of these occurrences attributed to goslings aged 5 to 10 days.

In Table 5, we observed that increased humidity had a significant impact on feather scores. Specifically, there were differences in feather scores between the Flat feeding and the Net-bed feeding group (*p* < 0.05).

### 3.6. Effects of Different Body Weight Compositions on Behavior and Feather Score of Goslings

For different body weight compositions (Figure 9A), it was found that the frequency of pecking in the Mixed weight group was significantly higher than that in the comparable weight group (*p* < 0.001). The frequency of pecking gradually increased from 3 days of age and reached a peak at 6 days of age (Figure 9B). Analysis of the feather scores revealed that the mixed-weight group had lower feather scores compared to the comparable weight group (*p* < 0.01) (Table 6). Body weight causes a change in the pecking sequence, with stronger goslings ranking higher, which also predisposes to aggressive feather pecking.

### 3.7. Effects of the Presence or Absence of Added Grass During Rearing on Gosling Behavior and Feather Scores

The results showed (Figure 10A) that the frequency of feather pecking behavior was significantly higher (*p* < 0.05) in goslings in the no grass-added group than in the grass-added group. Still, there was no significant difference in other behaviors. Notably, the frequency of cage pecking was higher in the goslings in the no grass-added group than in the added grass group, which also suggests that goslings’ feather pecking behavior is related to environmental enrichment. As shown in Figure 10B, the frequency of feather pecking in the no grass-added group was more than 60%, and at 8–10 days of age, almost 75% of the feather pecking was from the unadded grass group.

In Table 7, by analyzing the feather scores, it was found that the feather scores of goslings in the no grass-added group were significantly lower than those of goslings in the added grass group (*p* < 0.05).

## 4. Discussion

Throughout the entire observation period, the primary behaviors exhibited by goslings centered on foraging and drinking, which occurred almost concurrently and were not constrained by specific timeframes. Notably, feather-pecking behavior among goslings was more prevalent during intervals between feeding sessions. Research indicates that spontaneous activities and exploratory behaviors of laying hens serve as contributors to the induction of feather pecking [18]. Our observations and statistical analyses align with this finding, suggesting that goslings are more inclined to engage in feather-pecking when in an active state. Following feeding and drinking, goslings tend to collectively rest in the corners of the wire mesh bed, adopting a clustered lying posture that leads to feather wetting. Observational data confirm that before initiating feather-pecking, goslings actively seek out companions with wet backs or heads. Once feathers are pecked, they acquire a wet and stringy appearance, prompting continued feather-pecking by other goslings. This phenomenon aligns with the research findings reported by Eija, Cloutier, and Saskia [12,19,20].

Feather-pecking behavior can significantly degrade the feather quality of poultry, leading to increased heat dissipation requirements, food intake, and farming costs [21]. In severe cases, feather-pecking can cause skin and tissue damage and even result in the death of affected goslings [22,23]. Our histological section observations corroborate these findings, revealing varying degrees of skin damage, with large areas of the back skin of goslings exposed and accompanied by blood spots. Studies have also shown that poultry selects feather-pecking targets based on appearance characteristics, pecking individuals with artificially trimmed feathers significantly more frequently than those with intact feathers. Following trimming, feather-pecking and cannibalism behaviors rapidly spread within the group [24,25].

Furthermore, during the observation process, extreme feather-pecking individuals were identified, exhibiting obvious aggressive behaviors and continuously pecking other goslings. Hofmeyer termed these birds “feather-pecking experts”, who repeatedly pecked the feathers of their group mates [26]. In the present study, extreme feather-pecking behavior was predominantly observed at the age of 4–5 days. It is hypothesized that during this period, the hierarchical structure among goslings is established, resulting in the emergence of mutually aggressive feather-pecking behaviors within gosling groups [27].

With the rapid development of the poultry farming industry, many producers have adopted poor welfare conditions, such as the highest possible stocking density in production, as the economic benefits per unit area tend to be higher when the poultry stocking density is greater. However, poor welfare conditions can also hurt the production performance, behavior, and welfare of poultry [28]. In the current study, we observed that the occurrence of feather pecking behavior in goslings is influenced by stocking density, feeding conditions, body weight, and enrichment. Specifically, goslings reared under conditions of high density and high humidity exhibited increased feather pecking behavior, aligning with previous findings by Hughes and Yin [29]. Elevated stocking density exacerbated the mutual interactions among goslings, resulting in a significant increase in the frequency of feather pecking. Furthermore, the existing literature indicates that an increase in stocking density exacerbates feather damage among laying hens [30,31], a conclusion that finds support in our findings, which demonstrate decreased feather scores and pronounced feather damage within high-density gosling populations. Specifically, when the stocking density reaches excessively high levels, goslings exhibit heightened agitation, and the frequency of intragroup aggression rises, ultimately resulting in feather-pecking behavior [6].

Improper ventilation and humidity levels within poultry houses have been identified as factors contributing to feather pecking habits [7]. During our observations, we noted that the flat feeding created a humid environment, which caused the feathers of goslings to become damp. This, in turn, led to an increase in grooming and feather-pecking behaviors. Additionally, our study found that body weight differences among goslings also influenced feather pecking behavior. This may be attributed to variations in size and weight within the group, leading to the emergence of aggressive behaviors and the establishment of a hierarchical order. This is consistent with the results of Tahamtani’s study [32]; larger and heavier goslings typically occupy a higher position in the hierarchy and may have priority in accessing food and resting areas. Due to these size discrepancies, larger goslings were observed pecking smaller ones, prompting the latter to move around to avoid being pecked. Aggressive feather pecking was also noted in our observations, which was attributed to excessive stocking density and competition for food. In addition, Yngvesson and Keeling [33] found that chickens engaging in feather-pecking behavior had higher body weights than other chickens.

Environmental enrichment is a key factor contributing to feather pecking in goslings, and in previous studies, it has been found that enrichment of rearing conditions, such as the addition of materials and apomictic material, can reduce the incidence of feather pecking behavior [34,35,36]. In this study, we added pecking grass segments as an environmental enrichment factor to the goslings, and it was clearly found that the frequency of feather pecking behaviors of the goslings was reduced, and the cage pecking behaviors showed a tendency to decrease after the addition of grass. According to the redirected foraging hypothesis, poultry develop feather-pecking behavior when they do not have access to suitable foraging substrates such as wood chips, straw, or soil. Other types of enrichment, such as providing materials for sand bathing and novel (unfamiliar) objects, may be effective in reducing feather pecking. However, care should be taken when introducing novel objects, as some objects may cause fear and stress rather than have a positive effect on poultry [8,37].

Feather pecking behavior in goslings is a multifaceted animal behavior. By examining its relationship with the rearing environment, we can gain deeper insights into the underlying mechanisms and influencing factors of animal behavior. This research provides valuable references for future studies on similar behaviors in other animals and contributes positively to improving animal welfare and optimizing breeding practices.

## 5. Conclusions

This study has confirmed the significance of the brooding period in preventing feather pecking in goslings. In this study, high stocking density, floor rearing, low population uniformity, and limited environmental enrichment resulted in poor welfare conditions (including an increased frequency of abnormal behaviors such as feather pecking and lower feather scores). Feather pecking in goslings, with the back being the most commonly targeted area, can lead to the loss of back feathers and even skin damage, thereby affecting the health and growth of goslings.

## Figures and Tables

**Figure 1 animals-15-00616-f001:**
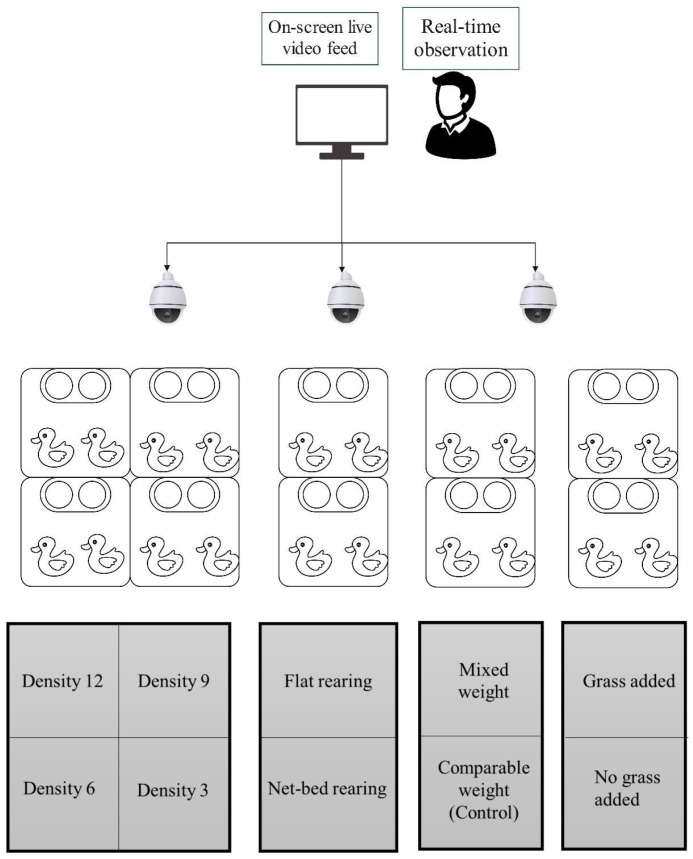
Schematic diagram of the rearing conditions for goslings based on the experimental strategy. A camera was placed above the net bed, and after the goslings fell off the plate, continuous observation and recording were made at 3 to 10 days of age, and various behaviors of goslings in different groups were observed through computer monitors and information on the occurrence of behaviors was recorded, as well as the degree and location of feather pecking behaviors.

**Figure 2 animals-15-00616-f002:**
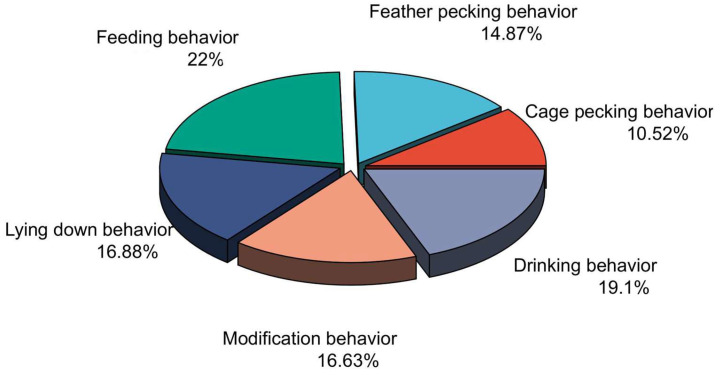
This figure summarizes the average frequency and percentage of occurrence for various behaviors, encompassing a total of goslings under all rearing conditions (N = 234).

**Figure 3 animals-15-00616-f003:**
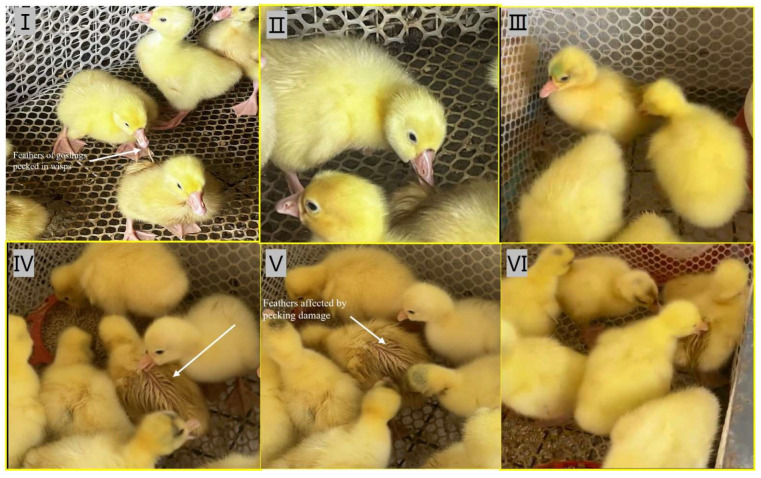
Feather pecking phenomenon in goslings. Figures (**I**–**III**) show that moist skin is particularly susceptible to feather pecking. As shown in Figures (**IV**–**VI**), when one gosling initiates feather pecking on its companion, it triggers a collective behavior of feather pecking among other goslings in the group.

**Figure 4 animals-15-00616-f004:**
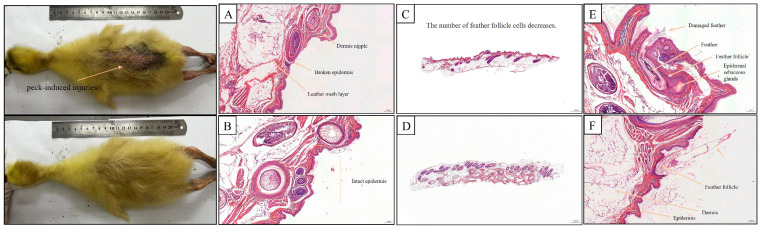
Analysis of dorsal skin tissues of goslings with intact and bareback feathers. (**A**,**C**,**E**) indicated pecked skin, while (**B**,**D**,**F**) indicated intact skin. (**A**) Skin epidermal breakage due to feather pecking. (**B**) Intact skin. (**C**,**D**) Comparison of feather follicle cell development and density between pecked and intact skin. (**E**,**F**) Comparison of feather follicle structure between pecked and intact skin.

**Figure 5 animals-15-00616-f005:**
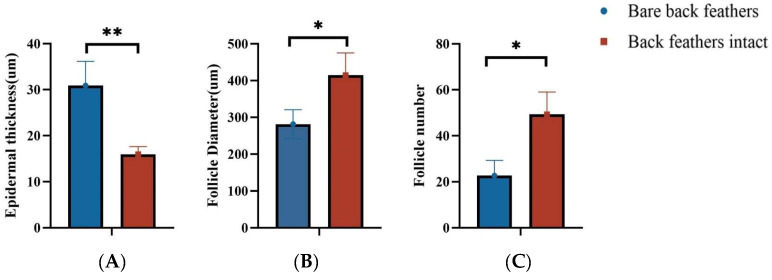
Comparison of goslings’ skin tissue characteristics: bareback feathers vs. intact back feathers. (**A**) Epidermal thickness measurements with bald back feathers and intact back feathers. (**B**) Feather follicle diameter from both groups. (**C**) Number of feather follicle cells. Each measurement was repeated three times for each sample. **, *p* < 0.01; *, *p* < 0.05.

**Figure 6 animals-15-00616-f006:**
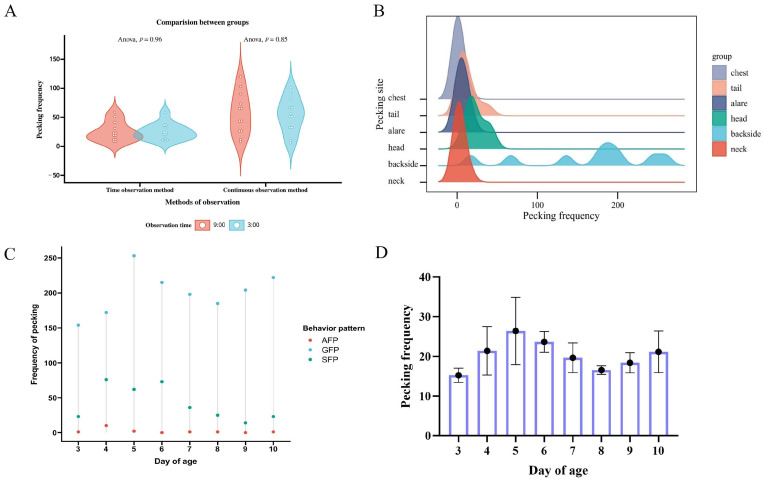
Analysis of gosling feather pecking behavior preferences and degree variations. (**A**) Comparison of the total frequency of feather pecking observed in goslings between morning (9 indicates 9:00 a.m.) and afternoon (3 indicates 3:00 p.m.). (**B**) Feather pecking site preference in goslings from 3 to 10 days of age. (**C**) Differences in the frequency of feather pecking in goslings at different days of age. GFP, gentle feather pecking; SFP, severe feather pecking; AFP, aggressive feather pecking. (**D**) Degree of feather pecking behaviors in goslings from 3 to 10 days of age (N = 234).

**Figure 7 animals-15-00616-f007:**
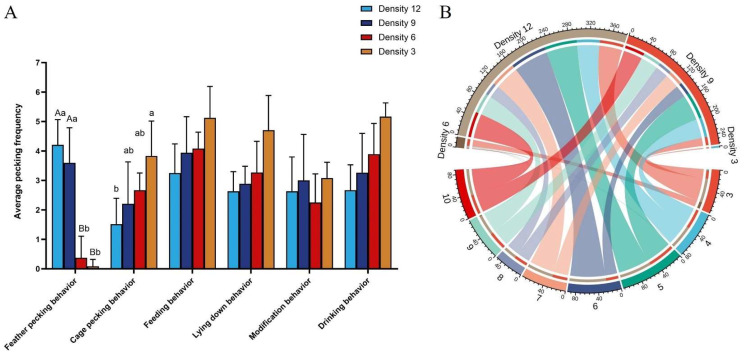
Different rearing densities on feather pecking behavior and feather coverage of goslings. (**A**) Differences in mean gosling behavior per gosling at different densities. No letter or the same lowercase letter means the difference is not significant (*p* > 0.05), different uppercase letters mean the difference is highly significant (*p* < 0.01), and different lowercase letters mean the difference is significant (*p* < 0.05), the same as below. (**B**) A chord diagram of the frequency of feather pecking at different ages based on data from different rearing densities. The outer circle represents the day of age and different groups, while the inner circle indicates the frequency of feather pecking. Color-coded ribbons were used to show the corresponding frequency between day age and groups (N = 90).

**Figure 8 animals-15-00616-f008:**
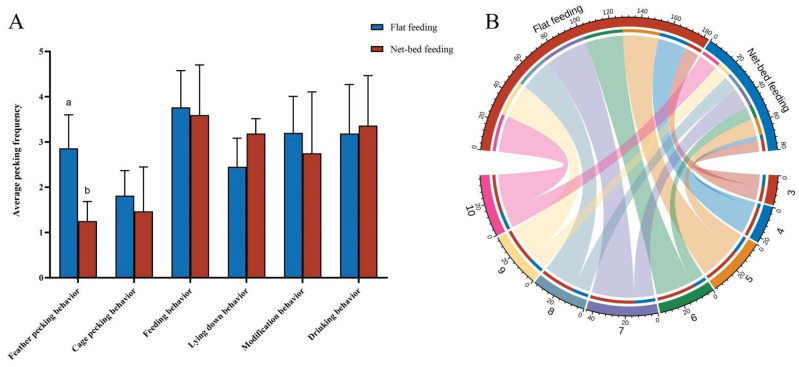
Effects of different rearing conditions on feather pecking behavior and feather coverage of goslings (N = 48). (**A**) Differences in mean feather pecking behavior of goslings under different rearing conditions. (**B**) A chord diagram of the frequency of feather pecking at different ages based on statistical data from different rearing conditions. The outer circle represents day age and different groups, while the inner circle indicates the frequency of feather pecking. Color-coded ribbons were used to show the corresponding frequency between day age and groups. Statistically significant differences are indicated by different letters.

**Figure 9 animals-15-00616-f009:**
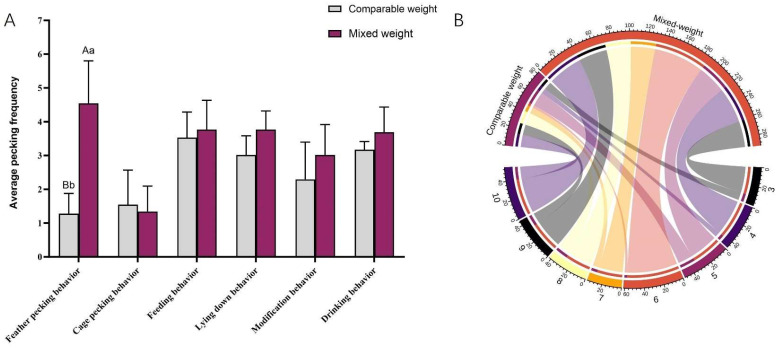
Analysis of different body weight compositions on behavior and feather coverage of goslings (N = 48). (**A**) Frequency of goslings’ behavior with different body weights. (**B**) A chord diagram of the frequency of feather pecking at different ages based on data from different genders and body weight compositions. Outer circles indicate age and different groups; inner circles indicate pecking frequency. Color-coded ribbons were used to show the corresponding frequency between day of age and groups. Statistically significant differences are indicated by different letters.

**Figure 10 animals-15-00616-f010:**
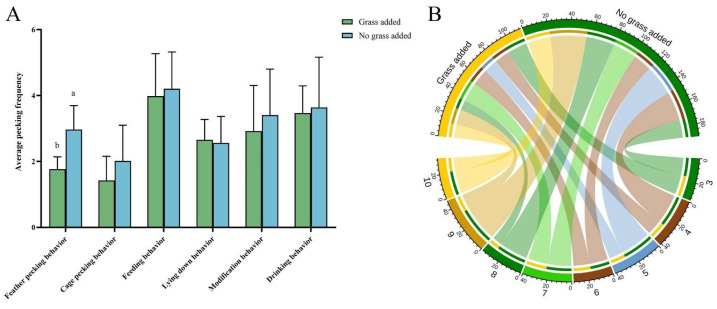
Analysis of whether or not to add grass on behavior and feather coverage of goslings (N = 48). (**A**) Frequency of goslings’ behavior. (**B**) A chord diagram of the frequency of feather pecking at different ages based on data on whether or not grass was added. Outer circles indicate age and different groups; inner circles indicate pecking frequency. Color-coded ribbons were used to show the corresponding frequency between day age and groups. Statistically significant differences are indicated by different letters.

**Table 1 animals-15-00616-t001:** Gosling behavior and its definitions.

Behavior Category	Behavior Definition
Feather pecking	Pecking or pulling on the feathers of other individuals, and in some cases, the pecked feathers are pulled off, and the pecked feathers are eaten.
Lying down	Chest on the ground without exhibiting other behaviors in the definition.
Feeding	Positioned next to a feeder with obvious feeding movements
Drinking	Beak was directed toward the drinker with obvious drinking motions.
Modification	Using the beak to gently rub, ruffle, or comb its feathers or use the toes to gently groom its wings
Cage pecking	Use the beak to peck at the surrounding netting material.

**Table 2 animals-15-00616-t002:** Feather scoring standard.

Score	Feathers	Part
5	Perfect Feathers	Chest, legs, back, tail and rump, wings, head, neck.
4	Damaged feathers, no skin areas exposed	
3	Bare area up to 3 cm × 3 cm	
2	Bare area between 3 cm × 3 cm and 5 cm × 5 cm	
1	Bare area greater than 5 cm × 5 cm	

**Table 3 animals-15-00616-t003:** Distinctions in the degree of feather pecking.

Behavior Pattern	Definition
Gentle feather pecking	Soft pecking on the plumage of other birds without pulling or removing feathers
Severe feather pecking	Forceful pulling, pecking, or removing feathers of other birds
Aggressive feather pecking	Forceful pecking from frontal, directed at the head of another bird.

**Table 4 animals-15-00616-t004:** Differences in feather scores under different environmental rearing densities.

Group	Score	Comparisons Between Groups	Z	*p*
Density 12	2 (1.25–3)	Density 12-Density 9	−0.104	0.917
		Density 12-Density 6	−2.839	0.005
		Density 12-Density 3	−2.413	0.016
Density 9	2 (2–3)	Density 9-Density 6	−2.606	0.009
Density 6	4 (3–4)	Density 9-Density 3	−2.267	0.023
Density 3	4 (4–5)	Density 6-Density 3	−0.195	0.845

**Table 5 animals-15-00616-t005:** Feather scores of goslings in different rearing conditions.

Group	Score	Z	*p*
Flat feeding	2 (2–2)	2.308	0.021
Net-bed feeding	2.5 (2–3)		

**Table 6 animals-15-00616-t006:** Differences in feather scores in feeding situations with different body weight compositions.

Group	Score	Z	*p*
Mixed weight	3 (2.25–3)	3.009	0.005
Comparable weight (Control)	4 (3.25–4)		

**Table 7 animals-15-00616-t007:** Differences in feather scores between rearing conditions Grass added and No grass added.

Group	Score	Z	*p*
Grass added	3 (3–4)	−2.348	0.019
No grass added	2 (2–3)		

## Data Availability

All data generated or analyzed during this study are included in this published paper.

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
