# Peer review of "Analysis of Risk Factors of Feather Pecking Injurious Behavior in Experimentally Raised Yangzhou Goslings in China"

_animals, 2025, doi:10.3390/ani15050616_

Round 1

Reviewer 1 Report

Comments and Suggestions for Authors

Dear authors

Both major and minor editing are suggested, as follows

-. You are suggested to edit the title. An alternative option for the title could be "Analysis of risk factors of feather pecking injurious behavior in experimentally raised Yangzhou goslings in China"

-. The term "hair follicle", which is widely used in this manuscript, is wrong, and should be removed and replaced by "feather follicle". 

-. L32-L33 Please rewrite the sentence "Significantly ... compared to low-density goslings".

-. L53 Please replace the term "ugly" for a suitable one.

-. L192 to L196 Please include arrows and/or asterisks in photos of Figure 3 to indicate the feather and skin lesions observed in pecked goslings.

-. L206 to L210 Please include arrows and/or asterisks in the gross and microscopic pictures of Figure 4. Please include a complete histopathologic description to differentiate affected and non-affected skin samples of studied goslings. 

-. L392 Please improve the length of the Conclusion section.

-. Listed references should follow the author guidelines of Animals.

-. Additional comment 1. Please include the reference "EFSA Panel on Animal Health and Animal Welfare (AHAW Panel) et al. Welfare of ducks, geese and quail on farm. EFSA Journal 2023, 21, 7992". Feather pecking is one of the geese welfare topics widely covered in this paper and could include a good option to enrich the Introduction, Discussion and Conclusions sections. Please follow the author guidelines of Animals to include and cite the named paper.

-. Additional comment 2. You provided valuable information describing the addition of grass to significantly reduce the incidence of feather pecking in the experimentally raised goslings. However, considering the addition of other materials such as pecking stones could be another strategy to reduce this injurious behaviour, and this could be included briefly here.

-. Additional comment 3. The production of down and feathers is one of the uses of farmed geese reared under commercial conditions. The authors could include a brief paragraph related to the potential benefits of identifying the risk factors of feather pecking at older age to reduce its negative impact on the down and feather industry.

Author Response

Dear reviewer #1:

  1. You are suggested to edit the title. An alternative option for the title could be "Analysis of risk factors of feather pecking injurious behavior in experimentally raised Yangzhou goslings in China"

Response: I would like to express my sincere gratitude for your professional guidance. Following your suggestion, we have made the necessary adjustments to the title of our manuscript. It now reads "Analysis of risk factors of feather pecking injurious behavior in experimentally raised Yangzhou goslings in China". Your input has been invaluable in refining the clarity and relevance of our research title, and we are confident that this change will enhance the overall presentation of our work. Thank you again for your time and expertise.

2.The term "hair follicle", which is widely used in this manuscript, is wrong, and should be removed and replaced by "feather follicle".

Response: Thank you for your careful review and valuable feedback. We have replaced all instances of "hair follicle" with "feather follicle" throughout the manuscript, including in Figure 5, and confirmed full consistency through a comprehensive review.

  1. L32-L33 Please rewrite the sentence "Significantly ... compared to low-density goslings".

Response: We gratefully appreciate your valuable comment.  Thank you for your suggestion.  We have revised as follows: "The higher feather pecking frequencies and poorer feather quality of goslings were observed under high-density conditions than those of lower-density environments. L30-32.

  1. -. L53 Please replace the term "ugly" for a suitable one.

Response: Thank you for pointing out this problem in this manuscript. We have changed "ugly in appearance" to "exhibit impaired plumage conditions". L53.

5.-. L192 to L196 Please include arrows and/or asterisks in photos of Figure 3 to indicate the feather and skin lesions observed in pecked goslings.

Response: Thank you for your valuable feedback. We have revised Figure 3 as requested by including arrows in the photos to clearly indicate the feather and skin lesions observed in pecked goslings. These markings will help readers easily identify the relevant areas of interest. The updated figure has been included in the revised manuscript. Please let us know if any further adjustments are needed. L200 to L204.

6.-. -. L206 to L210 Please include arrows and/or asterisks in the gross and microscopic pictures of Figure 4. Please include a complete histopathologic description to differentiate affected and non-affected skin samples of studied goslings.

Response: Thank you for your meticulous review and the specific suggestions provided. We have carefully revised and checked Figure 4 to ensure its quality and compliance you’re your requirements. If you have any further questions or need additional clarification, please feel free to let us know.  L213 to L216.

7.-. L392 Please improve the length of the Conclusion section.

Response: Thank you for your guidance and suggestions, we have made serious modifications:

This study has confirmed the significance of the brooding period in preventing feather pecking in goslings. In this study, high stocking density, floor rearing, low population uniformity, and limited environmental enrichment resulted in poor welfare conditions (including an increased frequency of abnormal behaviors such as feather pecking and lower feather scores). Feather pecking in goslings with the back being the most commonly targeted area can lead to the loss of back feathers and even skin damage, thereby affecting the health and growth of goslings. L411-417.

8.-. Listed references should follow the author guidelines of Animals.

Response: I truly appreciate your careful review and the valuable suggestion.  We have taken your advice very seriously. We have thoroughly examined and revised the reference list in our manuscript to ensure that all the listed references strictly adhere to the author guidelines of *Animals*. 

9.-. Additional comment 1. Please include the reference "EFSA Panel on Animal Health and Animal Welfare (AHAW Panel) et al. Welfare of ducks, geese and quail on farm. EFSA Journal 2023, 21, 7992". Feather pecking is one of the geese welfare topics widely covered in this paper and could include a good option to enrich the Introduction, Discussion and Conclusions sections. Please follow the author guidelines of Animals to include and cite the named paper.

Response: Thank you for your valuable suggestion. We have carefully revised the Introduction section and incorporated the reference "EFSA Panel on Animal Health and Animal Welfare (AHAW Panel) et al. Welfare of ducks, geese and quail on farm. EFSA Journal 2023, 21, 7992" as you recommended. [10]. This reference has proven to be extremely beneficial for our paper. It extensively covers the key topics related to geese welfare, which has significantly enriched the content of our Introduction, Discussion, and Conclusions sections. We followed the author's guidelines of *Animals* for the proper inclusion and citation of this paper, ensuring that it is seamlessly integrated into our manuscript. We believe that the addition of this reference enhances the scientific rigor and comprehensiveness of our work, providing a more solid foundation for our research on gosling feather pecking behavior.

10.-. Additional comment 2. You provided valuable information describing the addition of grass to significantly reduce the incidence of feather pecking in the experimentally raised goslings. However, considering the addition of other materials such as pecking stones could be another strategy to reduce this injurious behaviour, and this could be included briefly here.

Response: We truly appreciate your insightful comment. You're quite right that exploring additional materials to mitigate feather pecking is crucial. In our study, we focused on the significant effect of adding grass, which indeed demonstrated a remarkable reduction in the incidence of feather pecking among the experimentally raised goslings.  Regarding the suggestion of pecking stones, it is a promising avenue worth further investigation. Thank you again for your valuable feedback. It helps us to further refine and expand our work.

11.-. -. Additional comment 3. The production of down and feathers is one of the uses of farmed geese reared under commercial conditions. The authors could include a brief paragraph related to the potential benefits of identifying the risk factors of feather pecking at older age to reduce its negative impact on the down and feather industry.

Response: We are very grateful for your comment. In fact, feather-pecking behavior in older geese occurs less frequently, and unlike chickens, the beak of geese is blunt, which is not harmful to adult geese. Feather-pecking occurs mainly in the gosling stage, and is harmful to a certain extent. Thank you again for your valuable input.  It has provided us with a new perspective and direction for our work.

Reviewer 2 Report

Comments and Suggestions for Authors

Feather pecking is a common behavioral issue in goslings, characterized by one bird pecking at the feathers of another. This behavior can range from gentle nibbling to aggressive plucking, causing distress and potential injury to the recipient. Several factors can influence feather pecking in goslings, including genetics, social dynamics, and environmental conditions. However, the current study aims to reveal the differences in feather pecking behavior of goose goslings under different environments and the mitigation measures, to provide a certain theoretical basis for scientific selection and breeding, and to improve the productivity and economic benefits of waterfowl. It is an interesting article that is well written, but there are some comments & suggestions to improve this manuscript.

L.14: “these”: The letter “t” should be capitalized.

L.14: “these findings”: Which findings?

L.16: Also, the lighting intensity can impact.

L.28: “The victims”: Not well sound scientific!

L.27: “corresponding proportions”: Clarify.

L.23-28: Mention the stock density and the uniformity.

L.33: Correct to “high-density”.

L.38-39: I guess you mentioned that before! Please avoid repetition.

Is there a difference between feather pecking and cannibalism from your view?

“feather pecking in gosling has not been fully demonstrated”: Is there a critical reason for that?

Please add the lighting program in detail.

Did you determine the mortality percentages among groups?

What is the statistical design and model used for this experiment?

How did you differentiate between “severe feather pecking” and " aggressive one“?

“leaded to further pecking by conspecifics”: What is your explanation?

“goslings often pecked feathers during 4-5 days of age, most frequently directed at the back”: Are there deeper explanations for these remarks?

Has pecking been affected between morning and evening during the day (refer to records)?

Did you perform an economic study for this experiment?

Mention your novel results and expand your clarification, referring to previous literature.

Conclusion: Too general! Try to support it with a recommendation according to your findings.

References: Can be updated (last 10 years).

Table 5: Carefully check it.

Author Response

Dear reviewer #2:

1.L.14: “these”: The letter “t” should be capitalized.

Response: We have understood the issue and have already made the modification to "This study shows that". If there are any further concerns or if you need more detailed explanations, please let us know. L14.

2.L.14: “these findings”: Which findings?

Response: We have understood the issue and have already made the modification to "This study shows that". If there are any further concerns or if you need more detailed explanations, please let us know. L14.

  1. L.16: Also, the lighting intensity can impact.

Response: Thank you very much for your valuable suggestion regarding lighting conditions.     We greatly appreciate your insightful comment. You are correct that lighting intensity can significantly impact animal behavior. In our current study, the lighting procedure was so well developed that it was not set as a variable. Thank you again for your constructive feedback, which will undoubtedly enhance the quality of our future research.

  1. L.28: “The victims”: Not well sound scientific!

Response: We gratefully appreciate your valuable comment.  We have taken your advice to heart and promptly revised it to "The pecked gosling", which is much more precise and befitting the scientific context of our research. L26.

  1. L.27: “corresponding proportions”: Clarify.

Response: Thank you for bringing up the need to clarify the term "corresponding proportions".  We have carefully examined and revised the text as per your suggestion. As you correctly pointed out, the "corresponding proportions" refer to the percentages that each of the previously mentioned different pecking behaviors – namely gentle feather pecking (GFP), severe feather pecking (SFP), and aggressive pecking (AGP) – account for within the overall pecking behaviors observed. We've modified it as you suggested and changed it to "and the corresponding proportions". L25.

  1. L.23-28: Mention the stock density and the uniformity.

Response: Thank you for your comments, we have made the corresponding modification: risk factors of feather pecking injurious behavior were investigated including stocking density, rearing method, flock uniformity, environmental enrichment. L22-24.

  1. L.33: Correct to “high-density”.

Response: We sincerely appreciate your valuable feedback. In response to your suggestions, we have made corresponding revisions to the relevant content in the manuscript. L32

  1. L.38-39: I guess you mentioned that before! Please avoid repetition.

Response: Thank you for pointing out the issue of repetition on lines 38 - 39. We have carefully reviewed and made the necessary modifications.

  1. Is there a difference between feather pecking and cannibalism from your view?

Response: Thank you very much for your question. Cannibalism can occur either independently or after feasting (Blokhuis, 1998). There is no clear relationship between cannibalism and any kind of feather pecking, as there is no scientific support to suggest that cannibalism occurs to a higher degree or continues with feather pecking (Hughes and Duncan, 1972). We suggest that feather pecking and cannibalism are influenced by similar environmental conditions, but are not directly linked because cannibalism tends to occur prior to a large amount of feather pecking.

  1. “feather pecking in gosling has not been fully demonstrated”: Is there a critical reason for that?

Response: Thanks for your comments, we have made the following modifications: risk factors of feather pecking in goslings have not been fully demonstrated. Feather pecking by goslings has been reported previously but has not been studied in depth and more research has focused on chickens and ducks. L21.

  1. Please add the lighting program in detail.

Response: We appreciate your request regarding the lighting program details. To ensure the uniform growth of goslings, provide 23 - 24 hours of lighting time from 0 to 7 days of age. After 8 days of age, gradually transition from 24-hour lighting to using only natural light. L115-117.

  1. Did you determine the mortality percentages among groups?

Response: Thank you for your inquiry. In our experiment, there were no mortality cases among the gosling groups.

  1. What is the statistical design and model used for this experiment?

Response: Thank you for your question regarding the statistical design and model used in our experiment. Statistical Design: We employed a randomized controlled design for this study. The goslings were randomly assigned to different experimental groups to minimize the influence of confounding variables. This randomization process was carried out at the individual level to ensure that each gosling had an equal chance of being placed in any of the treatment or control groups. The experimental groups were defined based on different factors such as stocking density, and environmental enrichment (e.g., the presence or absence of grass or pecking stones).  Each group consisted of a sufficient number of goslings to provide adequate statistical power for detecting significant differences. We also incorporated replication within each group to account for individual variation and to enhance the reliability of our results.

Thank you for inquiring about the statistical analysis methods used in our experiment. In this study, we did not rely on a pre-specified statistical model. Instead, we followed a comprehensive approach to data analysis. Before any analysis, we conducted normality tests on all our data.   For the data that conformed to a normal distribution, we employed a one-way analysis of variance (ANOVA). This test allowed us to assess whether there were overall significant differences among the groups. After obtaining a significant result from the one-way ANOVA, we performed Duncan's multiple-range test. This post-hoc test helped us to precisely identify which specific groups differed significantly from one another. When using the one-way ANOVA, we presented the results as the mean ± standard deviation. On the other hand, for the data that did not follow a normal distribution, we used the Kruskal - Wallis test. This non - parametric test is suitable for comparing multiple independent groups when the normality assumption is violated. We reported the results of non-normally distributed data as the median (interquartile range). After a significant Kruskal-Wallis test, we carried out the Mann - Whitney test to make pairwise comparisons between groups. In all our statistical analyses, we considered a p-value less than 0.05 as statistically significant. This threshold enabled us to draw reliable conclusions about the differences between the experimental groups with the variables of interest, such as feather-pecking behavior and related influencing factors. We hope this detailed explanation clarifies our statistical analysis procedures.   If you have any further questions or need additional information, please feel free to let us know.

14.How did you differentiate between “severe feather pecking” and " aggressive one “?

Response: Aggressive pecking is usually a gosling that pecks at the head of another gosling. When pecking, the goslings do not tear the head feathers, but attack the other goslings in the form of pecking.  Severe pecking is when goslings tear the feathers of other goslings, or even tear off their feathers, we also refer to many similar studies in the literature for this identification standard.

[1] Zepp M, Louton H, Erhard M, et al. The influence of stocking density and enrichment on the occurrence of feather pecking and aggressive pecking behavior in laying hen chicks[J]. Journal of Veterinary Behavior, 2018:S1558787817301673.DOI:10.1016/j.jveb.2017.12.005.

Dong, Yiru (2019). Injurious pecking behavior of Pekin ducks on commercial farms: characteristics, development and duck welfare. Purdue University Graduate School. Thesis. https: //doi.org/ 10.25394/PGS.11323568.v1

[1] Genetic and Phenotypic Correlations Between Feather Pecking and Open-Field Response in Laying Hens at Two Different Ages[J]. Behavior Genetics, 2004, 34(4):407-415. DOI:10.1023/ B: BEGE.0000023646.46940.2d.

  1. “leaded to further pecking by conspecifics”: What is your explanation?

Response: When a gosling's feathers are pecked, areas such as its back will become wet due to the feather-pecking. This makes other goslings more inclined to peck at this gosling, rather than choosing those with healthy-looking feathers. This may be due to the imitative behavior of goslings or the existence of a hierarchical order among them.

  1. “goslings often pecked feathers during 4-5 days of age, most frequently directed at the back”: Are there deeper explanations for these remarks?

Response: Thank you for your insightful question regarding the observation that goslings often engage in feather-pecking behavior between 4 - 5 days of age, with the back being the most commonly targeted area. We offer the following in-depth explanations based on existing research and our understanding of avian behavior: Social Hierarchy Establishment. In this early stage, goslings begin to establish a social hierarchy within the group. For them, pecking may be a way to maintain dominance or test social boundaries. Compared with pecking more sensitive areas such as the head or eyes, pecking the back is a relatively non - aggressive initial approach. This allows them to initiate social contact and start to understand their position within the flock.

  1. Has pecking been affected between morning and evening during the day (refer to records)?

Response: We sincerely apologize for our oversight. Regarding your question, due to the heavy workload, our current experimental study only compared the differences between morning and afternoon. The results (Figure 6) show that there were no significant differences in pecking behavior among the groups during this period. In future studies, we will definitely make up for this deficiency by comparing the occurrence of pecking behavior between the day and the night.

  1. Did you perform an economic study for this experiment?

Response: Thank you for raising this important question. In our current experiment, we did not conduct a dedicated economic study. Our primary focus was on understanding the biological and behavioral aspects of feather pecking in goslings, such as identifying the factors influencing the behavior, its impact on the goslings' health, and potential mitigation strategies. However, we recognize the significant importance of economic implications in the context of poultry farming.    In future research, we plan to incorporate an economic analysis. We apologize for not including this aspect in the present study and are grateful for your guidance, which will help us to enhance the scope and practical relevance of our future research.

  1. Mention your novel results and expand your clarification, referring to previous literature.

Response: Thank you for your constructive feedback. I acknowledge that the previous conclusion was too general, and I have revised it by incorporating specific recommendations based on our research findings.

Revised Conclusion: This study has confirmed the significance of the brooding period in preventing feather pecking in goslings. In this study, high stocking density, floor rearing, low population uniformity, and limited environmental enrichment resulted in poor welfare conditions (including an increased frequency of abnormal behaviors such as feather pecking and lower feather scores). Feather pecking in goslings with the back being the most commonly targeted area can lead to the loss of back feathers and even skin damage, thereby affecting the health and growth of goslings.

  1. Conclusion: Too general! Try to support it with a recommendation according to your findings.

Response: Thank you for your constructive feedback. I acknowledge that the previous conclusion was too general, and I have revised it by incorporating specific recommendations based on our research findings.

Revised Conclusion: This study has confirmed the significance of the brooding period in preventing feather pecking in goslings. In this study, high stocking density, floor rearing, low population uniformity, and limited environmental enrichment resulted in poor welfare conditions (including an increased frequency of abnormal behaviors such as feather pecking and lower feather scores). Feather pecking in goslings with the back being the most commonly targeted area can lead to the loss of back feathers and even skin dam-age, thereby affecting the health and growth of goslings.

  1. References: Can be updated (last 10 years).

Response: Thank you very much for your suggestions. We have updated the references according to your comments

  1. Table 5: Carefully check it.

Group

Score

Comparisons between groups

Z

P

Density 12

2(1.25-3)

Density 12-Density 9

-0.104

0.917

Density 12-Density 6

-2.839

0.005

Density 12-Density 3

-2.413

0.016

Density 9

2(2-3)

Density 9-Density 6

-2.606

0.009

Density 6

4(3-4)

Density 9-Density 3

-2.267

0.023

Density 3

4(4-5)

Density 6-Density 3

-0.195

0.845

Response: Thank you for your attention to our work. We have carefully reviewed Table 5 and made revisions.

Reviewer 3 Report

Comments and Suggestions for Authors

The present study investigated the feather pecking behavior of gosling. The results showed that lower stocking density, higher population uniformity, and enrichment could reduce the prevalence of feather pecking. This study can provide important information on poultry industry. Following are some comments on this manuscript.

Title: The authors are suggested to clearly show the species of goose in the title.

Introduction: The authors are suggested to show the superiority of your research compared to other researches. Has the feather pecking behavior of gosling been investigated before? Or what are the results conducting on the other poultry species?

L104: The authors indicated that the experiment was set up with 3 replicates. You used a total of 70 goslings in this experiment. How can these be divided into 3 replicates?

It is not clear in the text.

L112-116: More details about the behavior observation are required. E.g. What kind of observation did you use? Did you record the duration of these behaviors? Or just record the frequency of these behaviors?

L114: Why did the authors choose to observe the behavior of gosling at these times? Is this the most active stage for gosling?

Figure 1: Was Figure 1 represent one replicate for the experiment? Based on the text, the number of goslings should be 78 (30+16+16+16). Again, how can these be divided into 3 replicates with 8 goslings in one group?

Table 4: The authors are suggested to show the videos about these feather pecking behaviors in supplementary files.

Figure 2: What was n= represent? Showing the percentage of behavior seems to be enough, there is no need to show n.

L190: Similar as above, why a total of 70 goslings were involved in the observation of behaviors?

Figure 7: N=30? What is the point of showing all sampling number of different groups? The authors should show the sampling number of each group. Similar with all figures.

Discussion: More discussion about why higher population uniformity could reduce the feather pecking behavior is required.

L382-385: The authors should elaborate more about how the enrichment mitigates the feather pecking behavior.

L386-390: The authors should discuss more about the practical implications of this study on poultry industry.

Author Response

Dear reviewer #3:

  1. Title: The authors are suggested to clearly show the species of goose in the title.

Response: I sincerely appreciate your valuable guidance. In response to your suggestion, we have revised the title of our manuscript.

2.Introduction: The authors are suggested to show the superiority of your research compared to other researches. Has the feather pecking behavior of gosling been investigated before? Or what are the results conducting on the other poultry species?

Response: Thank you for your constructive feedback. We have made revisions to the introduction section to highlight the superiority of our research. L56-64: As for the investigation of goslings' feather-pecking behavior, while there has been some research on the feather-pecking behavior of chickens and ducks as cited in references [6 - 8], the feather-pecking behavior of goslings has not been fully explored. Our study aims to fill this significant knowledge gap. Previous research on other poultry species has provided some insights into the effects related to injurious behaviors and group pressure during welfare assessments (as per reference [10]).

Our research stands out in several aspects. Firstly, we will comprehensively document the frequency of goslings' feather-pecking behavior, which has not been well-reported in previous literature. Secondly, we will identify the most commonly pecked body parts of goslings and the factors influencing such pecking, information that is currently scarce. We believe that these aspects demonstrate the superiority and novelty of our research, contributing to a more in-depth understanding of feather-pecking behavior in goslings and offering practical solutions for the poultry industry.

  1. L104: The authors indicated that the experiment was set up with 3 replicates. You used a total of 70 goslings in this experiment. How can these be divided into 3 replicates?

It is not clear in the text.

Response: We would like to express our sincere apologies for the confusion caused. In this text, we inaccurately stated the number of geese in a single experimental group. The correct information is that we had a total of 234 goslings, which were divided into three experimental groups, with each group containing 78 goslings. We understand the importance of accurate data reporting in research and will make sure to correct this error throughout the manuscript. Thank you for bringing this to our attention, and if you have any further questions or need additional clarification, please feel free to let us know.

  1. L112-116: More details about the behavior observation are required. E.g. What kind of observation did you use? Did you record the duration of these behaviors? Or just record the frequency of these behaviors?

Response: Thank you very much for your valuable feedback. We have taken note of your suggestions and will carry out the necessary modifications accordingly. The sampling method of all occurring events proposed by Altmann [17] was adopted. The data were expressed as the number of occurrences within the observation time. If a behavior occurs continuously within 5 seconds, it is counted as one observation result. Our data record the frequency of behavior. We are very sorry that due to the different duration of each behavior, we did not take into account the need to record when the behavior occurred. L117-120.

  1. L114: Why did the authors choose to observe the behavior of gosling at these times? Is this the most active stage for gosling?

Response: Thank you very much for your comments. Observation during the day allows for a clearer view of the occurrence of feather - pecking behavior in goslings, which improves the accuracy of the data. Moreover, goslings are in an active period before and after feeding, so we chose this time for observation and recording. Based on our observations, we also found that after being active for a while, goslings would lie down to rest for a long time and rarely exhibit active behaviors.

  1. Figure 1: Was Figure 1 represent one replicate for the experiment? Based on the text, the number of goslings should be 78 (30+16+16+16). Again, how can these be divided into 3 replicates with 8 goslings in one group?

Response: We truly apologize for the lack of clarity in our previous description.  As you pointed out, Figure 1 is indeed a repeated schematic diagram of the experiment, and 78 represents the number of goslings in one experimental group.  We deeply regret any confusion this has caused. To clarify further, the 78 goslings in the single experimental group were divided and assigned as follows: 30 were allocated to one subgroup, and then 16 each to the remaining three subgroups, making up the experimental setup of 30 + 16 + 16 + 16. We understand the significance of precision in scientific reporting and will meticulously check and modify the relevant parts of our manuscript to rectify this issue. L98.

  1. Table 4: The authors are suggested to show the videos about these feather pecking behaviors in supplementary files.

Response: Thank you for your insightful suggestion. We wholeheartedly agree that including videos of the feather-pecking behaviors in the supplementary files would significantly enhance the comprehensibility and impact of our study. We have retrieved the relevant video recordings of the feather-pecking behaviors. These videos were captured during our experiment with high-definition equipment to ensure clear visualization of the behaviors.

  1. Figure 2: What was n= represent? Showing the percentage of behavior seems to be enough, there is no need to show n.

Response: Thank you for your constructive feedback on Figure 2. We have carefully considered your suggestion and have made the necessary revisions. We have removed the “n=” indication from the figure. We believe this change enhances the clarity and simplicity of Figure 2, allowing readers to focus directly on the relevant behavioral percentages. We appreciate your guidance, which has helped us improve the quality of our manuscript. L196-199.

  1. L190: Similar as above, why a total of 70 goslings were involved in the observation of behaviors?

Response: We truly apologize for the lack of clarity in our previous description. We deeply regret any confusion this has caused. To clarify further, the 78 goslings in the single experimental group were divided and assigned as follows: 30 were allocated to one subgroup, and then 16 each to the remaining three subgroups, making up the experimental setup of 30 + 16 + 16 + 16. We understand the significance of precision in scientific reporting and will meticulously check and modify the relevant parts of our manuscript to rectify this issue.

  1. Figure 7: N=30? What is the point of showing all sampling number of different groups? The authors should show the sampling number of each group. Similar with all figures.

Response: We sincerely apologize for the error in Figure 7. It was a result of our oversight during the manuscript - preparation process. Initially, when we first conducted the statistical analysis, we adopted a sampling analysis approach. However, after review by our supervisor, the error was discovered. Therefore, we performed a reanalysis and modification. Unfortunately, in the rush of these changes, we failed to update the figure annotations, including the indication of the sampling number for each group in Figure 7. We understand the importance of accurate and detailed figure annotations in scientific research. We will immediately conduct a thorough review of all figures in the manuscript. For each figure, we will ensure that the sampling number of each group is clearly and correctly presented, following the appropriate format and requirements. Once again, we deeply regret this oversight and assure you that we will be more meticulous in future work to avoid similar mistakes. Thank you for your patience and for pointing out this issue to us.

  1. Discussion: More discussion about why higher population uniformity could reduce the feather pecking behavior is required.

Response: Thank you for your suggestions for the discussion section. We have added to the discussion to further explain the effect of population evenness on behavior occurrence. The additions are as follows: L389-391. In addition, Yngvesson and Keeling [30] found that chickens engaging in feather pecking behavior had higher body weights than other chickens.

  1. L382-385: The authors should elaborate more about how the enrichment mitigates the feather pecking behavior.

Response: Thank you for your suggestions for the discussion section. We added discussion to further explain the effect of environmental enrichment on behavior occurrence. The additions are as follows: L398-404: According to the redirected foraging hypothesis, poultry develop feather-pecking behavior when they do not have access to suitable foraging substrates such as wood chips, straw, or soil. Other types of enrichment, such as providing materials for sand bathing and novel (unfamiliar) objects, may be effective in reducing feather pecking; However, care should be taken when introducing novel objects, as some objects may cause fear and stress rather than have a positive effect on poultry [8,34].

  1. L386-390: The authors should discuss more about the practical implications of this study on poultry industry.

Response: Thank you for your valuable feedback on our manuscript. In response to your suggestions, we have made corresponding revisions. L360-364.

"With the rapid development of the poultry farming industry, many producers have adopted poor welfare conditions such as the highest possible stocking density in production, as the economic benefits per unit area tend to be higher when the poultry stocking density is greater. However, poor welfare conditions can also hurt the production performance, behavior, and welfare of poultry."

Round 2

Reviewer 1 Report

Comments and Suggestions for Authors

Dear authors

You addressed all suggestions and edits previously recommended. I have no additional comments to provide to you. 

Author Response

That's great to hear! Thank you so much for your time and careful review. I really appreciate your guidance throughout this process. If there's anything else in the future, please don't hesitate to let me know. Have a wonderful day! 

Reviewer 3 Report

Comments and Suggestions for Authors

The manuscript has been improved.

Author Response

(The authors gave the same response as above.)
